# The Utility of Mitochondrial Detection Methods Applied as an Additional Tool for the Differentiation of Renal Cell Tumors

**DOI:** 10.3390/diagnostics13142319

**Published:** 2023-07-09

**Authors:** Gorana Nikolic, Maja Zivotic, Sanja Cirovic, Sanja Despotovic, Dusko Dundjerovic, Sanja Radojevic Skodric

**Affiliations:** 1Institute of Pathology, Faculty of Medicine, University of Belgrade, 11000 Belgrade, Serbia; maja.zivotic@med.bg.ac.rs (M.Z.); sanja.cirovic@med.bg.ac.rs (S.C.); 2Institute for Histology and Embryology “Aleksandar Đ. Kostić”, Faculty of Medicine, University of Belgrade, 11000 Belgrade, Serbia; sanja.despotovic@med.bg.ac.rs

**Keywords:** renal cell tumors, renal cell carcinomas, immunohistochemistry, immunofluorescence, mitochondria, electron microscopy

## Abstract

The precise differentiation of renal cell tumors (RCTs) is sometimes hard to achieve using standard imaging and histopathological methods, especially for those with eosinophilic features. It has been suggested that the vast overabundance of mitochondria, as a well-known hallmark of eosinophilic cytoplasm, and could be a characteristic of distinct tumor types with opposing clinical outcomes. Thus, we intended to explore the associations between mitochondrial distribution patterns in different RCTs, including 43 cell renal cell carcinomas (ccRCCs), 15 papillary renal cell carcinomas (pRCCs), 20 chromophobe renal cell carcinomas (chRCCs), and 18 renal oncocytomas (ROs). Tumor samples were stained with two anti-mitochondrial antibodies (mitochondrial antibody Ab-2, clone MTC02; prohibitin, II-14-10, MA5-12858), applying immunohistochemistry and immunofluorescence to define mitochondrial distribution patterns (coarse scanty, moderate granular, and diffuse granular). Our results revealed significantly different expression patterns among the investigated RCTs (*p* < 0.001). The majority of ccRCCs exhibited coarse scanty mitochondrial staining, while all chRCCs had moderate granular expression. Nevertheless, all ROs, all pRCCs, and two cases of ccRCC presenting with higher nuclear grade and eosinophilic cytoplasm had diffuse granular mitochondrial expression. Moreover, with increased distribution of mitochondria, the intensity of staining was higher (*p* < 0.001). Here we present a strategy that utilizes fast and easy mitochondrial detection to differentiate RO from chRCC, as well as other eosinophilic variants of RCC with high accuracy.

## 1. Introduction

The World Health Organization (WHO) in 2022 proposed a new classification of adult renal cell tumors based on histological findings, immunophenotypes, molecular findings, and patient outcomes [1]. The most common types of renal tumors are clear cell renal cell carcinoma (ccRCC), papillary renal cell carcinoma (pRCC), chromophobe renal cell carcinoma (chRCC), and renal oncocytoma (RO). The diagnosis of two specific subtypes, renal oncocytomas (RO) and chromophobe renal cell carcinoma (chRCC), however, remains a challenge due to overlapping histopathological features and the same origin [2]. Namely, the differentiation between chRCC and RO using standard hematoxylin and eosin-stained sections is particularly difficult and often requires ancillary studies. In addition to RO and chRCC, a category of ‘other oncocytic tumors’ has been introduced, representing a heterogeneous tumor group not classifiable as the aforementioned entities or other tumor types with eosinophilic features. These entities are not specific pathohistological diagnoses and represent tumor category requiring a wide spectrum of additional research, influencing patient treatment and outcomes [3].

Different immunohistochemical stains have been used to differentiate kidney tumors. It is known that both cytokeratin (CKAE1/AE3) and vimentin show positive staining in ccRCC and pRCC, while vimentin shows negative staining in chRCC and RO. Furthermore, in addition to CK and vimentin, CK7 shows positive staining in pRCC, chRCC, as well as ccRCC with papillary features. RO shows absent or focal positivity of CK7. Although CK7 positivity in most of the cases distinguishes chRCC from RO, studies have shown that the eosinophilic variant of chRCC overlaps with RO in the expression of CK7 [4], especially on needle biopsies [5,6]. Another useful antibody that has been frequently used over the past two decades and appears to be useful in the differential diagnosis of kidney tumors is an anti-mitochondrial antibody, particularly if molecular analyses are not available. Distribution of anti-mitochondrial staining in ccRCC could be granular but is usually scanty, while immunoexpression in pRCC is intense, granular, and diffuse. Immunoexpression in most ROs show intense, diffuse, granular staining compared with the perinuclear, moderate, granular staining pattern in chRCC [7,8].

In addition to immunohistochemical staining, transmission electron microscopy (TEM) provides valuable ultrastructural information in kidney tumor differentiation, mostly the distinction between RO and chRCC [9,10]. The tumor is considered RO if the tumor cells contain abundant mitochondria with lamellar cristae, occupying the whole cytoplasm. The most prominent TEM features of chRCC are multiple membrane-bound microvesicles and less prominent mitochondria containing tubular cristae, which are dominantly perinuclearly positioned, compared to RO [11]. pRCC analyzed by TEM also shows numerous mitochondria [12]. Nevertheless, ccRCC explored by TEM revealed few mitochondria per cell or even a complete absence [13].

As is known, immunofluorescence is widely and routinely utilized for the evaluation of non-tumor kidney diseases, both in frozen and paraffin-embedded tissue [14,15]. However, its possibilities have not been fully explored in kidney tumor diagnosis. Considering that the most prominent ultrastructural differences between the most common types of renal cell tumors rely on the abundance and distribution pattern of mitochondria when explored by TEM, we aimed to investigate whether immunohistochemistry and immunofluorescence applying anti-mitochondrial antibodies could be used as an additional technique for the adequate diagnosis of these renal tumors. 

## 2. Materials and Methods

### 2.1. Tissue Samples

The current study included 96 patients diagnosed with renal cell tumors. Among them, 43 samples belonged to cases of clear cell renal cell carcinoma (ccRCC), 15 samples were diagnosed as papillary renal cell carcinoma (pRCC), 20 samples were chromophobe renal cell carcinoma (chRCC), and 18 samples were cases of renal oncocytoma (RO). All diagnoses were reviewed by two experienced uropathologists and were based on hematoxylin and eosin (H&E), histochemical (HC), and immunohistochemical (IHC) slides using light microscopy (LM). All kidney samples, using the aforementioned analyses, were explored on slides from formalin-fixed paraffin-embedded (FFPE) blocks and contained tumor and adjacent non-tumor kidney parenchyma. The material was collected from the archives of the Pathology Department, Clinic of Urology, University Clinical Center of Serbia, Belgrade. The Ethics Committee of the Clinic of Urology of Clinical Center of Serbia approved collecting the samples from the archive and carrying out the study (Application Ref: 0152/20, dated 4 March 2020). The study presented here was conducted following all ethical standards laid down in the 1964 Declaration of Helsinki. Since the study was carried out retrospectively, according to our ethical guidelines, informed consent of the patients was not required.

### 2.2. Clinical Patient Data and Gross Tumor Examination

All clinical patient data (gender, age, tumor side) and data obtained during gross pathohistological examination (tumor dimensions) were collected from medical records and pathohistological reports.

### 2.3. Routine Light Microscopy Tissue Analyses

Paraffin blocks were cut into 4 µm thick sections and afterward stained routinely with H&E. Depending on the tumor morphology and differential diagnosis, additional HC (Halle colloid iron) and/or IHC (CAIX, CKAE1/AE3, CK7, CD117, CD10, vimentin, AMACR, RCC) were performed to obtain a final pathohistological diagnosis. All diagnoses and recommendations for nuclear grading were applied according to the new WHO classification, especially for pRCCs, which are now classified as low and high nuclear grades. The nuclear grading system according to WHO/ISUP was also implemented. Slides were analyzed using an Olympus BX51 light microscope, an Olympus C5060A-ADU digital camera, and analySIS 5.0 software (Soft Imaging System, Olympus, Tokyo, Japan). 

### 2.4. Immunohistochemical Staining and Analysis

All kidney samples were selected and IHC staining was performed with two antibodies in order to detect different epitopes specifically found in mitochondria. The first antibody recognized a 60 kDa non-glycosylated protein component of the mitochondria found in normal and malignant human cells (mitochondrial antibody Ab-2 (clone MTC02, Thermo Scientific, Waltham, MA, USA, dilution 1:100). The second antibody, prohibitin (II-14-10, MA5-12858, Thermo Scientific, dilution 1:50), was introduced in order to explore an evolutionarily conserved protein located in the inner membrane of the mitochondria. Tumor sections (4 µm thick) were deparaffinized and rehydrated. Antigen retrieval procedures were performed using citrate buffer (pH 6.0, 20 min in the microwave). The primary antibodies were incubated for 1 h at room temperature. Sections were then treated with the EnVision^TM^ Detection System (DAKO, Glostrup, Germany), using 3,3’-diaminobenzidine or 3-amino-9-ethyl carbazole as the substrate, and counterstained with hematoxylin. Negative controls were performed by omitting the first antibody and afterward stained using the EnVision^TM^ method. Three pathologists reviewed all slides for the presence or absence of mitochondrial expression in the tumor samples. The slides were evaluated using an Olympus BX51 light microscope, an Olympus C5060A-ADU digital camera, and analySIS 5.0 software (Soft Imaging System, Olympus).

The IHC stained slides were analyzed with regard to distribution patterns and intensity of mitochondrial staining. The distribution patterns were defined as coarse scanty (score 1), moderate granular (score 2), and diffuse granular (score 3) cytoplasmic staining, while intensity was defined as mild (1), moderate (2), and strong (3). Finally, the IHC score comprised a sum of these two scores.

### 2.5. Immunofluorescence Staining and Analysis

The same FFPE specimens that were stained by IHC with both mitochondrial antibodies (mitochondrial antibody Ab-2, clone MTC02, and prohibitin, II-14-10, MA5-12858), were also used in the IF analysis to determine the sensitivity of the above-mentioned antibodies to these types of staining. For that purpose, 4 µm thick FFPE sections were treated as previously described [16]. The FFPE sections were then treated with the antigen retrieval procedure applying citrate buffer of pH 6.0, as described in the IHC analysis section (20 min). Afterward, the slides were incubated overnight with the primary mouse monoclonal mitochondrial antibody Ab-2 (clone MTC02, Thermo Scientific, dilution 1:100), followed by the secondary anti-mouse IgG-Alexa 488 (ab150113, Abcam, 1:100) for 45 min at room temperature. The cell nuclei were identified by counterstaining with 4,6-diamino-2-phenylindolyldihydrochloride (DAPI; 1 µg/mL Sigma-Aldrich, Darmstadt, Germany). Negative controls were performed in all experiments by omitting the first antibody. In each staining experiment, cross-binding of the secondary fluorescence-labeled antibody was controlled on specific control slides on which the first and/ or second antibodies were omitted. Tissue sections were mounted with Fluoro Preserve Reagent (Calboichem, Darmstadt, Germany). All of the slides were analyzed by confocal microscopy using a Leica TCS SP2 microscope (Leica Microsystem, Wetzlar, Germany) with an x63/NA 1.49 oil immersion objective and Leica Confocal Software. For the FITC-conjugated secondary antibodies, the excitation wavelength was 488 nm, while emission was detected through a long-pass filter above 520 nm.

### 2.6. Transmission Electron Microscopy 

Transmission electron microscopy (TEM) analysis was performed on kidney samples in paraffin blocks that were cut into smaller pieces, approximately 2 × 2 mm in size, deparaffinized in xylol (2 min, four times), rehydrated in graded ethanol (100%-2 changes, 96%, 70%, 50%, each change 5 min) to distilled water, and rinsed in phosphate buffer solution for 5 min. The samples were re-fixed in 3% glutaraldehyde in cacodylate buffer, post-fixed in 1% OsO4 and 4,8% uranyl-acetate, dehydrated in graded ethanol, and embedded in Epoxy medium, according to the standard procedure [17]. The ultrathin sections were stained with uranyl acetate and lead citrate and examined using a Transmission Electron Microscope (TEM) (Morgagni 268D FEI, Hillsboro, OR, USA).

## 3. Results

### 3.1. Expression Pattern of Anti-Mitochondrial Antibody in Renal Cell Tumors—Description of Patient Cohort 

The study included 96 patients diagnosed with the most frequent pathohistological types of renal cell tumors. Their clinical and pathohistological characteristics were analyzed in three different groups, based on the IHC staining pattern of the anti-mitochondrial antibody (Table 1). The staining patterns were defined as coarse scanty, moderate granular, and diffuse granular cytoplasmic staining. The majority of males (51.9%) had coarse scanty cytoplasmic staining, while females had a similar distribution of all three staining patterns (Table 1). Patients were of similar ages in all groups, without a significant difference (Table 1). Renal cell tumors were surgically removed from both left and right kidneys, and the tumor side did not influence the staining pattern, as shown in Table 1. There were no clinical data on tumor side for two patients; therefore, they were excluded from the statistical analysis. The average values of the largest tumor dimensions were not statistically different within the groups and were usually around 60 mm (Table 1). The largest tumor was diagnosed as ccRCC with a maximal dimension of 200 mm, and the largest diameter of the smallest one, also diagnosed as ccRCC, was 13 mm. Statistically significant different expression patterns were observed in various pathohistological renal cell tumor types (Table 1). Among them, ccRCC was mostly diagnosed with coarse scanty mitochondrial staining detected by IHC, with the exception of two cases found with diffuse granular staining patterns of moderate intensity. These two ccRCC cases presented with higher nuclear grades (*p* = 0.005), as shown in Table 1. All pRCC and RO had diffuse granular cytoplasmic staining, while chRCC was exclusively diagnosed with moderate granular staining. These differences among pathohistological types were statistically significant (*p* < 0.001). Staging of tumor disease was also explored concerning the expression patterns of mitochondrial IHC staining (Table 1); however, a significant difference was not observed (*p* = 0.764). Overall, it was observed that IHC staining intensity was increased in groups with moderate and diffuse granular patterns (*p* < 0.001), as presented in Table 1.

### 3.2. Visualization of Mitochondria in Non-Tumor Kidney Samples

The non-tumor adjacent kidney parenchyma is shown in Figure 1A. By applying immunohistochemistry and immunofluorescence, mitochondria were detected both in proximal and distal tubules, showing a variable, intense, granular staining pattern with the anti-mitochondrial antibody, while the glomeruli were completely negative for staining (Figure 1B,C). Moreover, TEM microphotography (Figure 1D) confirmed a higher number of mitochondria in the proximal tubule, which could be compared with the stronger staining intensity obtained by IHC (Figure 1B) and IF (Figure 1C) analyses.

### 3.3. Clear Cell Renal Cell Carcinoma—ccRCC

Figure 2 illustrates the ccRCC morphology of low (Figure 2A) and high nuclear grades (Figure 2E). Using the anti-mitochondrial antibody, coarse scanty IHC staining was visible in the majority of ccRCCs. The detected granules were dispersed and randomly distributed, as shown in Figure 2B,F. Similarly, when applying immunofluorescence, very weak signals were detected in IF images (Figure 2C,G). Morphologically, ccRCC with lower nuclear grade and clear cytoplasm (Figure 2A) exhibited fewer mitochondria detected by TEM (Figure 2D) compared to ccRCC with higher nuclear grade and slightly eosinophilic cytoplasm (Figure 2E), which showed an increased number of mitochondria (Figure 2F–H). Despite increased numbers of mitochondria in ccRCC with higher nuclear grade (Figure 2H), precise ultrastructural analyses could not be performed because of poor TEM image resolution, representing a consequence of inadequate tissue fixation since they were made from FFPE samples.

### 3.4. Papillary Renal Cell Carcinoma—pRCC

All pRCC samples expressed diffuse granular IHC staining patterns. However, depending on the pRCC morphology, we noticed slight differences in mitochondrial antibody expression patterns. Indeed, low-grade pRCC (defined as type 1 according to the previous edition of the WHO classification), illustrated in Figure 3A, had a diffuse granular uniform pattern throughout the whole cytoplasm both in IHC (Figure 3B) and IF (Figure 3C) slides. On the other hand, high-grade pRCC (defined as type 2 according to the previous edition of the WHO classification), illustrated in Figure 3E, had a diffuse granular pattern with slight apical accentuation (Figure 3F,G). Morphologically, voluminous eosinophilic cells visible on optical microscopy (Figure 3E) exhibited prominent, tightly packed mitochondria with apical accentuation in TEM analysis (Figure 3H), while low-grade pRCC had uniform, dispersed mitochondrial distribution with lower tendency to accumulate on the apical cytoplasmic side (Figure 3D).

### 3.5. Chromophobe Renal Cell Carcinoma—chRCC

The morphology of chRCC could be classic, with predominant pale tumor cells (Figure 4A), and could also consider eosinophilic cell type (Figure 4E). The chRCC types included in our study mainly belonged to the eosinophilic type, and only three cases had both pale and eosinophilic components. The IHC analysis of chRCC showed moderate granular cytoplasmic staining, localized mostly along the cell membrane. Some dispersed granular staining was detected in the cytoplasm, but areas around the nuclei remained unstained, illustrating a perinuclear halo (Figure 4B,F). Numerous granules were also detected by IF with heterogeneous IF signal intensity, but to a lesser extent compared to IHC staining (Figure 4C,G). Scattered mitochondria were predominantly localized on the periphery of the cell (along the cell membrane), but in general, they had a moderate granular staining pattern. A prominent cell membrane was visible morphologically (Figure 4A,E), as well as by TEM (Figure 4H). TEM analyses also confirmed the distribution of mitochondria and enabled the detection of some intracytoplasmic microvesicles of unknown origin presented as paler structures, specifically found in chRCC.

### 3.6. Renal Oncocytoma—RO 

The staining in RO was diffuse granular with strong intensity (Figure 5B). The same pattern was confirmed by IF staining (Figure 5C). Bearing in mind that electron microscopy confirmed that RO is rich in mitochondria (Figure 5D), our immunostaining analyses, both IF and IHC, verified that anti-mitochondrial staining could be specific and sensitive for RO. Furthermore, it was also confirmed that “oncocytic” cell morphology (Figure 5A) was a clear consequence of numerous mitochondria diffusely localized within the cytoplasm. 

### 3.7. Summary of Immunohistochemistry and Immunofluorescence Applying Anti-Mitochondrial Antibody in Renal Cell Tumors

All relevant results, based on the IHC and IF findings, are summarized in Table 2. In addition, negative controls for IHC-stained mitochondrial samples are shown in Figure 6 to confirm antibody specificity. Additionally, Figure 6 also includes positively stained slides with another mitochondrial antibody (prohibitin monoclonal antibody (II-14-10, MA5-12858). Prohibitin is an evolutionarily conserved protein located in the inner membrane of the mitochondria. In contrast, initial staining with mitochondrial antibody Ab2 (clone MTC02) recognizes the non-glycosylated protein components of the mitochondria. Thus, the two different antibodies are specific for the detection of two different epitopes in mitochondria. Despite the use of two different antibodies, our results showed agreement. Regardless of the mitochondrial antibody used, the staining results were comparable. 

## 4. Discussion

It is well known that classic ccRCC and pRCC have specific morphological features that enable easier distinction. The main challenges remain the eosinophilic variants of RCT due to their morphological overlapping, especially using only H&E slides. Therefore, additional diagnostic methods, such as IHC, HC staining, TEM, and molecular profiling, are in use. In addition to these methods being time-consuming and requiring different tissue fixation, they also have additional limitations. 

Considering the quality of IHC staining from FFPE samples, IHC staining highly depends on the adequacy of the preanalytical phase, such as tissue fixation procedures, as well as the establishment of staining protocol, balancing between peroxidase and protein blockade, antigen retrieval procedures, antibody clones, their dilutions, and the visualization of antigen–antibody reactions; thus, the results of IHC staining and their interpretation can sometimes be challenging for pathologists [18,19]. This is especially true when highly specialized pathohistological laboratories receive samples previously fixed and paraffin-embedded in other institutions since there is no single way to standardize and establish unique protocols [19]. Due to non-standardized and variable FFPE tissue preparation among laboratories, such as different fixative solutions, duration of paraffin baths, etc., further microscopic slide staining procedures could lead to false negativity or even unspecific positivity in IHC [19]. For example, HC staining such as Hale’s colloidal iron, frequently used for differentiation between RO and chRCC, fails to stain adequately [11,20]. Moreover, renal neoplasms with focal mucin-like changes, such as oncocytomas and papillary renal neoplasms, could be mild to moderately stained with Hale’s colloidal iron. Thus, RCT with extensive chromophobe cell-like features may pose a differential diagnostic problem with chRCC when the pathohistological examination is based on HC staining [21]. In that case, TEM is a gold standard for distinguishing these types of tumors. According to the literature data, tumor cells of RO are filled with uniform and round mitochondria. They are closely packed in the cytoplasm and their size is larger than those in chRCC. In addition to the mitochondria, scattered microvesicles are also present and their numbers vary from cell to cell. They are either absent or very sparse, contrary to chRCC. In chRCC, the number and shape of mitochondria depend on the number of microvesicles. Their distribution can be diffuse or localized in the peripheral cytoplasm [12,22]. However, TEM is not widely used for daily practice, considering the requirements of specially trained human resources (technicians and pathologists) along with expensive equipment and high depreciation costs [23]. It is also important to use an adequate fixation procedure for TEM since all required fine ultrastructural details such as mitochondria cristae and microvesicles could not be interpreted on TEM slides from FFPE tissues, as we noticed in our results. Here, we were only able to detect mitochondrial presence, their number, localization, and partially their shape, but we could not see all of the mentioned important details of their structure, as authors reported in RCT samples properly fixed for TEM analysis [12]. Indeed, in order to use TEM for diagnostic purposes in uropathology, a piece of kidney tumor sample should be stored in adequate fixative.

In addition to standard H&E and HC slides and a routinely used wide-panel spectrum of IHC antibodies and TEM analysis, the authors explored the utility of different mitochondrial antibodies, considering that the main ultrastructural differentiation of RTCs is based on this cellular organelle differences in major RCT types. Therefore, Mete et al. used an anti-mitochondrial antibody (AMA, NeoMarkers), reporting that the pattern of immunoreactivity was mostly diffuse in RO, while diffuse but peripherally accentuated in most chRCCs; thus, they proposed very high sensitivity (96%) and specificity (94%) of AMA staining for differentiation of chRCCs from ROs [7]. Similarly, Kuroda et al. detected mitochondria by MIA (Biogenex) staining and showed strong and diffuse positivity in oncocytic chRCC, without the description of the staining pattern [8]. In the same regard, Tickoo et al. described diffuse and fine granular positivity of antimitochondrial antibody 113-1 in all examined ROs and peripheral accentuation of coarse cytoplasmic granules in chRCCs. Another eosinophilic and granular type of tumor mostly showed irregular coarsely granular and diffuse staining [24]. In our study, we used two mitochondrial antibodies (mitochondrial antibody Ab-2, clone MTC02; prohibitin monoclonal antibody, II-14-10, MA5-12858), obtaining diffuse granular cytoplasmic staining in RO and moderate granularity and mostly periphery-localized staining among chRCCs. In ccRCC, coarse scanty cytoplasmic staining was observed in most cases. However, we also detected increased mitochondrial expression in the ccRCCs with higher nuclear grades and with eosinophilic cytoplasm. All pRCC samples expressed diffuse granular IHC staining patterns. Our results are in accordance with the literature data, so regardless of the clone in use, mitochondrial antibodies can be included in a standard IHC panel for renal neoplasms since they could provide additional hints for precise diagnosis. 

Here we showed an agreement between the IF results and the results obtained by IHC, which described the mitochondrial staining patterns in different types of RCTs. The observed differences, with regard to mitochondrial presence and distribution, were confirmed by TEM. All of these data suggest that the application of a single antibody could assist in a quick and precise diagnosis, avoiding implementation of the broad and expensive IHC panel currently used for the differentiation of RCTs. IF analysis is not widely used in routine diagnostics of renal neoplasms since the preparation and evaluation of immunofluorescence images require expertise. However, the trainee does not require much time [25]. Moreover, the standard H&E ex tempore pathohistological examination is not always accurate and precise, considering that fresh-frozen tissue samples do not have preserved morphology. Thus, this type of unfixed fresh-frozen tissue sample enables high sensitivity and high specificity in IF staining procedures, allowing clear positioning, the use of multiple antibodies, and beautiful staining [26,27,28,29,30], leading to fast and precise identification of mitochondria and thereby enabling the quick establishment of a precise diagnosis. Quick establishment of a precise diagnosis is mainly useful for differentiation when clinical, radiological, and morphological characteristics could be similar in benign and malignant tumors, such as in renal oncocytoma and chromophobe renal cell carcinoma.

## 5. Conclusions

The application of immunostaining using anti-mitochondrial antibodies can assist in the fast and easy distinction of renal oncocytoma from chromophobe renal cell carcinoma and other eosinophilic variants of RCC with high accuracy, since it has been established that the vast overabundance of mitochondria is a well-known hallmark of eosinophilic cytoplasm [31]. Therefore, it can be used as an additional tool when a diagnosis needs to be obtained quickly. In that regard, since immunofluorescence analysis can be performed on fresh-frozen tissue samples used for ex tempore analysis during preoperative and intraoperative procedures, a fast decision could be made about the best surgical protocol for the patient, balancing the benefits of total or partial nephrectomy.

## Figures and Tables

**Figure 1 diagnostics-13-02319-f001:**
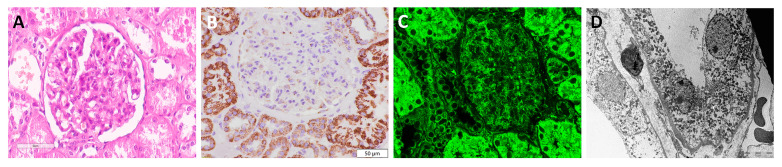
Mitochondrial visualization in normal kidney. (**A**) H&E×40, (**B**) IHC×40 (*Mitochondria antibody Ab-2, clone MTC02*), (**C**) IF×60, and (**D**) TEM×28,000 microphotograph of mitochondria; distribution in normal human kidney, illustrating a higher density of mitochondria in proximal tubuli than distal tubuli.

**Figure 2 diagnostics-13-02319-f002:**
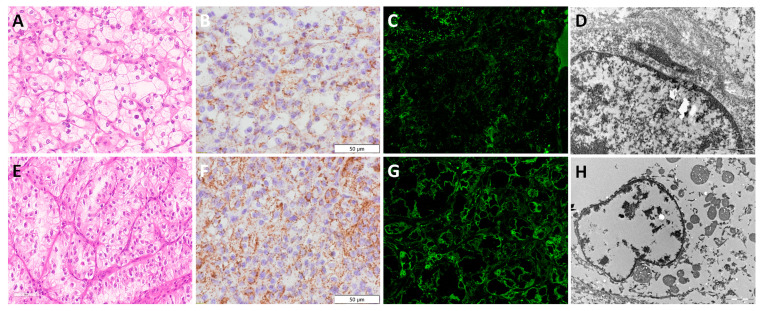
Mitochondrial visualization in low and high nuclear grade ccRCC. Low-grade ccRCC (**A**–**D**), (**A**) H&E×40, (**B**) IHC×40 (*Mitochondria antibody Ab-2*, *clone MTC02*), (**C**) IF×60, and (**D**) TEM×22,000 microphotographs of mitochondrial distribution, illustrating the morphology and decreased numbers of mitochondria compared to high-grade ccRCC (**E**–**H**), (**E**) H&E×40, (**F**) IHC×40 (*Mitochondria antibody Ab-2*, *clone MTC02*), (**G**) IF×60, and (**H**) TEM×7100 microphotographs, representing increased numbers of mitochondria.

**Figure 3 diagnostics-13-02319-f003:**
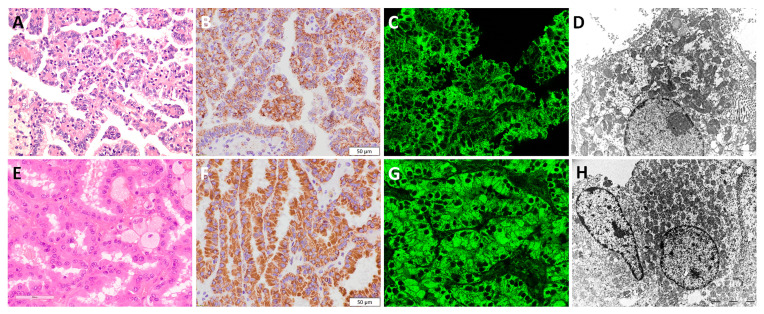
Mitochondrial visualization in low- and high-grade pRCC. Low-grade pRCC (**A**–**D**), (**A**) H&E×40, (**B**) IHC×40 (*Mitochondria antibody Ab-2*, *clone MTC02*), (**C**) IF, and (**D**) TEM×7100 microphotographs of mitochondrial distribution, illustrating the diffuse mitochondrial staining but with less prominent apical accentuation detected in high-grade pRCC (**E**–**H**), (**E**) H&E, (**F**) IHC (*Mitochondria antibody Ab-2*, *clone MTC02*), (**G**) IF, and (**H**) TEM×4400 microphotographs of mitochondrial distribution.

**Figure 4 diagnostics-13-02319-f004:**
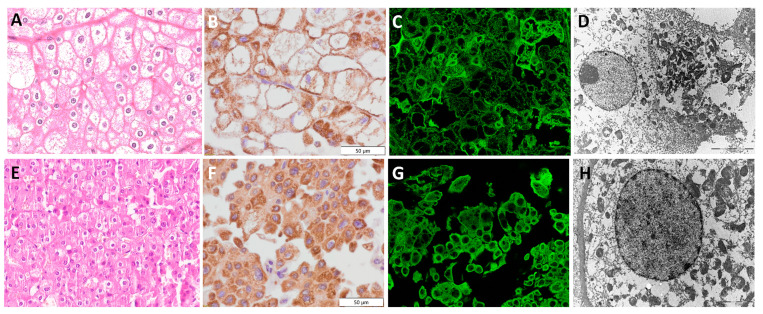
Mitochondrial visualization in classic chRCC (**A**–**D**) and eosinophilic chRCC (**E**–**H**). Classic chRCC, (**A**) H&E×40, (**B**) IHC×40 (*Mitochondria antibody Ab-2*, *clone MTC02*), (**C**) IF×60, and (**D**) TEM×4400 microphotographs of mitochondrial distribution, illustrating the mitochondria, which were less prominent compared to eosinophilic chRCC, (**E**) H&E×40, (**F**) IHC×40 (*Mitochondria antibody Ab-2*, *clone MTC02*), (**G**) IF×60, and (**H**) TEM×7100 microphotographs.

**Figure 5 diagnostics-13-02319-f005:**
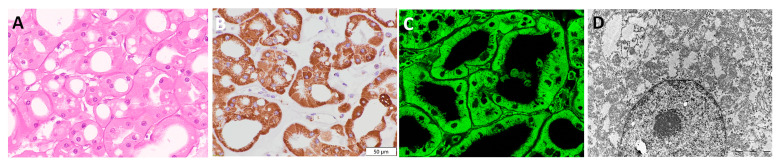
Mitochondrial visualization in RO. (**A**) H&E×40, (**B**) IHC×40 *(Mitochondria antibody Ab-2*, *clone MTC02*), (**C**) IF×60, and (**D**) TEM×8900 microphotograph of mitochondrial distribution, illustrating the prominent mitochondria tightly packed throughout the whole cytoplasm, without apical accentuation.

**Figure 6 diagnostics-13-02319-f006:**
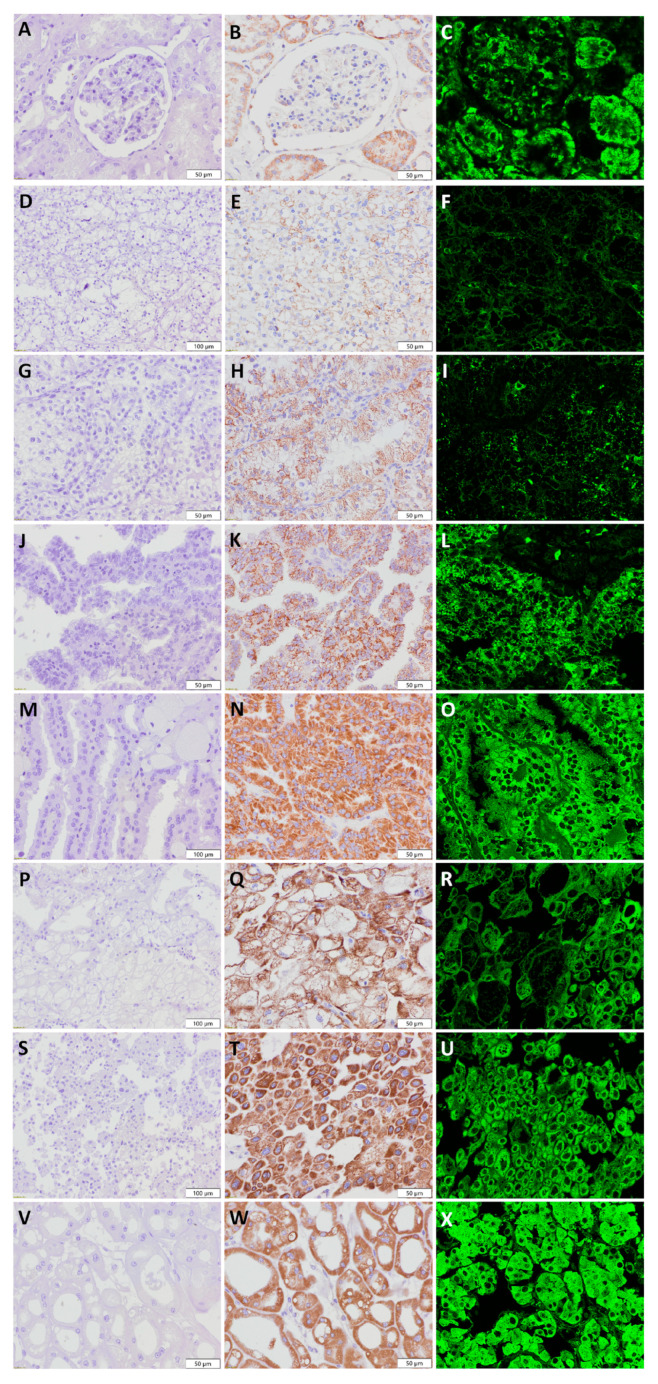
Validation staining with prohibitin antibody. Mitochondrial visualization in: normal kidney (**A**–**C**), (**A**) negative control×40, (**B**) IHC×40, and (**C**) IF×60; low-grade ccRCC (**D**–**F**), (**D**) negative control×40, (**E**) IHC×40, and (**F**) IF×60; high-grade ccRCC (**G**–**I**), (**G**) negative control×40, (**H**) IHC×40, and (**I**) IF×60; low-grade pRCC (**J**–**L**), (**J**) negative control×40, (**K**) IHC×40, and (**L**) IF×60; high-grade pRCC (**M**–**O**), (**M**) negative control×40, (**N**) IHC×40, and (**O**) IF×60; chRCC-classic variant (**P**–**R**), (**P**) negative control×40, (**Q**) IHC×40, and (**R**) IF×60; chRCC-eosinophilic variant (**S**–**U**), (**S**) negative control×40, (**T**) IHC×40, and (**U**) IF×60; RO (**V**–**X**), (**V**) negative control×40, (**W**) IHC×40, and (**X**) IF×60.

**Table 1 diagnostics-13-02319-t001:** Clinical and pathological characteristics of patient cohorts with three different mitochondrial staining patterns.

Clinical and Pathological Characteristics		Distribution Pattern of Mitochondrial Staining			*p*
		Coarse Scanty	Moderate Granular	Diffuse Granular	
Gender	male	28 (51.9 %)	5 (9.3 %)	21 (38.9 %)	*p* = 0.005
n (%)	female	13 (31.0%)	15 (35.7 %)	14 (33.3 %)	
Age (average age ± SD)		60.9 ± 9.0	61.6 ± 8.3	61.2 ± 10.3	*p* = 0.960
Side	left	23 (43.4 %)	9 (17.0 %)	21 (39.6%)	*p* = 0.478
n (%)	right	17 (41.5 %)	11 (26.8%)	13 (31.7%)	
Highest tumor dimension (average mm ± SD)		64.8 ± 36.5	59.8 ±26.2	63.9 ± 34.2	*p* = 0.865
Tumor type	Clear cell RCC	41 (95.3 %)	0 (0.0 %)	2 (4.7 %)	*p* < 0.001
n (%)	Papillary RCC	0 (0.0 %)	0 (0.0 %)	15 (100 %)	
	Chromophobe RCC	0 (0.0 %)	20 (100 %)	0 (0.0 %)	
	Oncocytoma	0 (0.0 %)	0 (0.0 %)	18 (100 %)	
Nuclear grade- ccRCC #	Low grade (NG I. II)	30 (100 %)	0 (0.0%)	0 (0.0%)	*p* = 0.005
n (%)	High grade (NG III. IV)	11 (84.6 %)	0 (0.0%)	2 (15.4 %)	
T staging	T1a	8 (44.4 %)	4 (22.2 %)	6 (33.3 %)	*p* = 0.764
n (%)	T1b	20 (71.4 %)	7 (25.0 %)	1 (3.6 %)	
	T2a	5 (35.7 %)	5 (35.7 %)	4 (28.6 %)	
	T2b	0 (0.0%)	1 (50.0 %)	1 (50.0%)	
	T3a	6 (50.0 %)	3 (25.0 %)	3 (25.0 %)	
	T3b	1 (100.0 %)	0 (0.0%)	0 (0.0%)	
	T4	1 (100.0 %)	0 (0.0%)	0 (0.0%)	
Expression intensity of anti-mitochondrial antibody	mild moderate	41 (95.3 %) 0 (0.0%)	0 (0.0%) 0 (0.0%)	0 (0.0%) 2 (4.7 %)	*p* < 0.001
	strong	0 (0.0%)	20 (37.7 %)	33 (62.3 %)	

#—nuclear grading system according to WHO/ISUP for ccRCC.

**Table 2 diagnostics-13-02319-t002:** Immunohistochemistry and immunofluorescence results applying anti-mitochondrial antibody in renal cell tumors.

Type of RCTs		IHC and IF Findings	IHC Score (Mean ± SD)	*p*
ccRCC	low NG	Mild intensity with coarse scanty cytoplasmic distribution pattern	2.14 ± 0.64	<0.001
	high NG	Moderate intensity with diffuse granular cytoplasmic staining		
pRCC	low-grade	Strong intensity with diffuse granular staining throughout the whole cytoplasm, with slight apical accentuation	5.67 ± 0.49	
	high-grade	Strong intensity with diffuse granular staining throughout the whole cytoplasm, with prominent apical accentuation		
chRCC	classic pattern	Strong intensity with mild to moderate granular distribution staining pattern, localized mostly along the cell membrane	4.80 ± 0.41	
	eosinophilic pattern	Strong intensity with moderate granular distribution staining pattern, localized mostly along the cell membrane		
RO	eosinophilic pattern	Strong intensity with diffuse granular staining throughout the whole cytoplasm	6.00 ± 0.00	

## Data Availability

All data are provided within the manuscript.

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
