# Peer review of "The Utility of Mitochondrial Detection Methods Applied as an Additional Tool for the Differentiation of Renal Cell Tumors"

_diagnostics, 2023, doi:10.3390/diagnostics13142319_

Round 1

Reviewer 1 Report

The authors G. Nikolic et al. describe in the present manuscript entitled “The utility of mitochondria detection methods applied as an additional tool for the differentiation of renal cell tumours” a well-founded study on a the application of an anti-mitochondria antibody which can be used as an additional fast diagnosis tool in the field of renal cell tumours. This new tool can easily distinct between the different types of tumours which is sometimes hard with the well established pathological methods.

As I don’t see any obstacles in using this additional tool for the best surgical protocol for the patient, I would recommend to publish this study as it is.

Author Response

Dear reviewer, thank you very much for your suggestion for publication our manuscript without revision. However, we made some changes according to requirements of other reviewers, which are clearly indicated by Track Changes in the revised version of the manuscript.

Reviewer 2 Report

The manuscript titled, “The utility of mitochondria detection methods applied as an additional tool for the differentiation of renal cell tumors” lay focus on an interesting topic but has some serious lacunae as follows:

1.       The IHC data needs to be quantified and a correlation of the staining with patient tumor cancer progression has to be analyzed.

2.       The immunofluorescence data cannot be accepted. It is not at all implied whether the green fluorescent signal is indicating the mitochondria or something else. High resolution confocal images have to be provided for these staining images.

3.       Why only one single mitochondrial antibody was used? The authors need to show a cohesive finding involving different mitochondrial compartments.

4.       The TEM images are of extremely poor resolution such that the mitochondria cannot be clearly seen or analyzed. Need better TEM images with quantification.

Author Response

Reviewer 2.

1."The IHC data needs to be quantified and a correlation of the staining with patient tumor cancer progression has to be analyzed. "

Response:

Here we quantified IHC data, which is presented as a score of the IHC staining distribution pattern and expression intensity of the anti-mitochondria antibody. These data are illustrated in Table 2. However, regarding patient tumor cancer progression, we couldn’t perform analysis (despite a large cohort) because the follow-up period was too short and because of the pandemic situation over the past three years, it was hard to distinguish disease-specific causes of death.

"The immunofluorescence data cannot be accepted. It is not at all implied whether the green fluorescent signal is indicating the mitochondria or something else. High-resolution confocal images have to be provided for these staining images."

Response: 

Dear reviewer, according to your suggestion, we performed confocal microscopy, preparing all samples in the same way as before, as well as high-resolution confocal images.

  1. "Why only one single mitochondrial antibody was used? The authors need to show a cohesive finding involving different mitochondrial compartments."

Response: 

By applying the antibody used in our study, we performed a comparative analysis with results obtained by other research groups and accordingly showed overlapping features regarding the clone we used. The mentioned researchers also used single antibody clones for their work. All these details are mentioned in the Discussion.

  1. "The TEM images are of extremely poor resolution, such that the mitochondria cannot be clearly seen or analyzed. Need better TEM images with quantification."

Response

We are aware of the poor TEM image resolution, which is mainly a consequence of TEM slide preparation from FFPE tissue samples that were improperly fixed. This type of analysis has already been discussed and explained in terms of the advantages and disadvantages of FFPE tissue using TEM. Since this was a retrospective study, we did not routinely collect fresh, frozen, unfixed tissue for TEM considering that TEM is not applied in routine diagnostic work in the uropathology field. Thus, we discussed it: "It is also important to use an adequate fixation procedure for TEM, since all required fine ultrastructural details such as mitochondrial cristae and microvesicles could not be interpreted on TEM slides from FFPE tissues, as we noticed in our results. Here, we were only able to detect mitochondrial presence, their number, localization, and partially their shape, but we could not see all the important details of their structure, as authors found in RCTs properly fixed for TEM analysis [12]. Indeed, in order to use TEM for diagnostic purposes in uropathology, a piece of kidney tumor sample should be stored in an adequate fixative."

Reviewer 3 Report

In this manuscript, Nikolic et al. developed a new classification method for renal cell tumor diagnostics. This method uses an anti-mitochondria antibody to accurately detect mitochondria amount, which serves as a quantitative metric for tumor classification. This work is suitable for this journal, and I have a few comments:

1, How specific is the mitochondria antibody? Is it known which sequences the antibody binds?

2, What’s the biological basis for distinguishing tumor types using mitochondria? More discussion is needed to address this.

3, What’s the cost of mitochondria staining compared to the current standard methods?

Author Response

Reviewer 3 (round 1):

In this manuscript, Nikolic et al. developed a new classification method for renal cell tumor diagnostics. This method uses an anti-mitochondria antibody to accurately detect mitochondria amount, which serves as a quantitative metric for tumor classification. This work is suitable for this journal, and I have a few comments:

1, How specific is the mitochondria antibody? Is it known which sequences the antibody binds?

From available manufacturer data sheet, under Product Specific Information it is written: “This antibody recognizes a 60kDa non-glycosylated protein component of mitochondria found in normal and malignant human cells.” In the same document there is Figure showing the double IF staining of antibody used in our study together with MitoTracker® Red CMXRos (Product # M7512) mitochondria marker. Thus Panel d represents the merged image showing mitochondria colocalization. It is indicated below:

Moreover, according to the suggestion of Reviewer 2, we stained the same samples with a different antibody (Prohibitin Monoclonal Antibody (II-14-10) (MA5-12858)) in order to confirm the conclusions. Prohibitin is an evolutionarily conserved protein located in the inner membrane of mitochondria. On the other hand, initial staining with mitochondria-Ab2 (clone MTC02) recognizes non-glycosylated protein components of mitochondria. Thus, two different antibodies are specific for the detection of two different epitopes in mitochondria. 

2, What’s the biological basis for distinguishing tumor types using mitochondria? More discussion is needed to address this.

In the current revised manuscript version, we included in the discussion section the following sentence: “Quick establishment of the precise diagnosis is mainly useful for differentiation when clinical, radiological and morphological characteristics could be similar in benign and malignant tumours, such as renal oncocytoma and chromophobe renal cell carcinoma.” 

3, What’s the cost of mitochondria staining compared to the current standard methods?

Compared to the standard methods with implementation of broad IHC panel, the advantage of mitochondria staining is much more chipper. Thus we included in the Discussion the following comments: Here we showed an agreement between the IF results and the results obtained by IHC, which describe mitochondria staining patterns in different types of RCTs. The observed differences, with regard to mitochondria presence and distribution, were confirmed by TEM. All these data suggest that the application of a single antibody could fast assist in a precise diagnosis, avoiding also the implementation of broad and expensive currently used IHC panel for differentiation of RCTs.”

Reviewer 4 Report

Differential diagnosis of renal cell tumors (RCTs) is usually easy even with conventional hematoxylin & eosin (H&E) staining, but differentiation among clear cell renal cell carcinoma (RCC) (ccRCC), granular cell variant, chromophobe RCC (chRCC), and renal oncocytoma (RO) may be sometimes challenging. In this study, the authors investigated the utility of mitochondria detection in differential diagnosis of those RCTs by immunohistochemistry (IHC) and immunofluorescence (IF).

The purpose is clearly described, and background information is well presented. However, I have some concerns in data analyses and their presentation. Furthermore, discussion and ensuing conclusion should also be modified.

1. In Table 1, distribution pattern of mitochondria staining is classified as coarse scanty, moderate scanty, or diffuse granular. Almost all ccRCC was coarse scanty, all papillary RCC (pRCC) was diffuse granular, all chRCC was moderate granular, and all RO was diffuse granular. Expression intensity is dichotomized in mild to moderate or strong. On the other hand, in the Discussion, it is also described that diffuse intense fine granular cytoplasmic staining in RO, intensive with coarse granularity in chRCC, and diffuse disperse granular in pRCC (Lines 287-288). As such, there are differences in description in between Table 1 and in Discussion, which are very confusing. I recommend description of IHC patterns united between them.

In addition, description that diffuse uniform pattern throughout the whole cytoplasm (pRCC, low grade) and intense granular pattern apically accentuated (pRCC, high grade) (Lines 292-294) should also be revised.

2. Moderate granular pattern and diffuse granular pattern of mitochondria distribution cannot be well discriminated with Figure 1E, G, I. So, they should be replaced with images with higher magnification and higher resolution. In addition, diffuse uniform pattern throughout the whole cytoplasm (pRCC, low grade) and intense granular pattern apically accentuated (pRCC, high grade) should be presented with high resolution images.

3. In Table 1, a total number of cases of left and right sides is 94, fewer than 96.

4. In Table 1, please specify the nuclear grading system of ccRCC.

5. In Figure 2, images of transmission electron microsope (TEM) should be explained in captions.

6. In Discussion, it is described that mitochondria in RO is larger than that of chRCC (Lines 257-258). However, as far as I see Fig. 2 L and O, it is not the case.

7. Please explain what scattered microvesicles in TEM mean.

8. In Discussion, it is described that all these results (IF and IHC) correlated with the TEM visualized mitochondria (Lines 299-300). In Figure 1, IHC and IF signals are weaker in pRCC than in chRCC and RO, but in Figure 2, it looks that density of mitochondria is higher in pRCC than in chRCC and RO. There is an inconsistency.

9. There are two kinds of cells, eosinophilic cells and pale cells in chRCC. I presume that authors analyzed eosinophilic cells. What are the results of mitochondria expression in pale cells?

10. In conclusion, it is described that IF using anti-mitochondria antibody can assist in fast and easy distinction of RO from chRCC and other eosinophilic variant of RCC with high accuracy. However, I can find little discussion that can result in a conclusion that IF is more useful than IHC.

11. In the Discussion, I recommend a table that specifies the differential characteristics of IHC and IF of these RCTs.

I think that “beside” may be misused. Furthermore, there are many other minor errors in English and I strongly recommend that this manuscript undergo extensive English editing by native speakers.

Author Response

Dear Reviewer, 

We are really sorry for huge mistake, unintentionally made during submission of the revised manuscript together with the responses to all reviewers. Indeed, all your suggestions were responded point by point, but unfortunately were sent to reviewer number three instead to you. Again, we are very sorry for the inconveniency and thus send you all the responses below. 

1."In Table 1, distribution pattern of mitochondria staining is classified as coarse scanty, moderate scanty, or diffuse granular. Almost all ccRCC was coarse scanty, all papillary RCC (pRCC) was diffuse granular, all chRCC was moderate granular, and all RO was diffuse granular. Expression intensity is dichotomized in mild to moderate or strong. On the other hand, in the Discussion, it is also described that diffuse intense fine granular cytoplasmic staining in RO, intensive with coarse granularity in chRCC, and diffuse disperse granular in pRCC (Lines 287-288). As such, there are differences in description in between Table 1 and in Discussion, which are very confusing. I recommend description of IHC patterns united between them. n addition, description that diffuse uniform pattern throughout the whole cytoplasm (pRCC, low grade) and intense granular pattern apically accentuated (pRCC, high grade) (Lines 292-294) should also be revised."

Response:

In order to maintain the uniform terminology within the whole manuscript, we revised the text in the Results and Discussion (track changes) according to the nomenclature used in the Table 1 headings. With regard to your last comment,

Lines 292-294, we adjusted the description in the Results section:

"All pRCC samples expressed diffuse granular IHC staining patterns. Although the current 5th edition of the WHO classification of urinary and male genital tumours does not differ between pRCC as type 1 and type 2, we noticed slight differences in mitochondria antibody expression patterns depending on the morphology. Indeed, low-grade pRCC (defined as type 1 according to the previous edition of WHO classification) had a diffuse, uniform granular uniform pattern throughout the whole cytoplasm, while high-grade pRCC (defined as type 2 according to the previous edition of WHO classification) had a diffuse granular pattern with slight apical accentuation." In addition, the Discussion section was also modified. Thus, these sentences were deleted, but the changes were included and sufficiently explained in the Results.

  1. "Moderate granular pattern and diffuse granular pattern of mitochondria distribution cannot be well discriminated with Figure 1E, G, I. So, they should be replaced with images with higher magnification and higher resolution. In addition, diffuse uniform pattern throughout the whole cytoplasm (pRCC, low grade) and intense granular pattern apically accentuated (pRCC, high grade) should be presented with high resolution images."

Response:

Figure 1 is now modified with the higher magnification and higher resolution images. Moreover, the new Figure 3 contains two morphologically different types of pRCC. These two types were previously classified as pRCC type 1, and pRCC type 2 (WHO, 4th edition, 2016), reclassified as low and high grade pRCC in 5th edition of WHO classification (2022). Considering that pRCC type 2 morphologically contains more eosinophilic cells with abundant cytoplasm, contrary to pRCC type 1, the expression of anti-mitochondria antibody was slightly different, which we tried to explain with apically accentuated expression in high grade pRCC (pRCC, type 2).

3." In Table 1, a total number of cases of left and right sides is 94, fewer than 96."

Response:

In Table 1, the total number of RCTs divided by sides is indeed 94, because in two cases we did not receive clinical data. In the revised version of the manuscript, this information is included in the text "For two patients, there was no clinical data on tumour side; therefore, two out of 96 cases were excluded from statistical analysis".

4."In Table 1, please specify the nuclear grading system of ccRCC"

Response:

The nuclear grading system for ccRCC was used according to the WHO/ISUP scheme, and this is implemented in the Table 1 legend. Moreover, the following sentences are now implemented in Methodology Section 2.3: "All diagnosis and recommendations for nuclear grading were applied according to the new WHO classification, especially for pRCC, which are now classified as low and high grade. A nuclear grading system according to WHO/ISUP is also implemented."

  1. "In Figure 2, images of transmission electron microsope (TEM) should be explained in captions."

Response:

Under the Figures legend, we shortly described TEM finding, in investigated RCTs, while in the text of the Results, all details are sufficiently explained.

  1. "In Discussion, it is described that mitochondria in RO is larger than that of chRCC (Lines 257-258). However, as far as I see Fig. 2 L and O, it is not the case. "

Response:

In this part, we discussed literature data with further comparison to our findings without insights into the precise ultrastructural findings because our TEM results were based on FFPE tissue and, accordingly, were not properly fixed for precise TEM analysis. Therefore, we mentioned at the begging of this discussion part that these ultrastructural differences correspond to literature data and do not represent our findings. The following sentences are slightly adjusted according to your comments:"According to literature data, tumour cells of RO are filled with uniform and round mitochondria. They are closely packed in the cytoplasm, and their size is larger than those in chRCC. Beside mitochondria, scattered microvesicles are also present, and their number varies from cell to cell. They are either absent or very sparse, contrary to chRCC. In chRCC, the number and shape of mitochondria depend on the number of microvesicles. Their distribution can be diffuse or localized in the peripheral cytoplasm [12, 22]. However, TEM is not widely used for daily practice, considering the requirements of specially trained human resources (technicians and pathologists), along with the expensive equipment and high depreciation costs [23]. It is also important to use an adequate fixation procedure for TEM since all required fine ultrastructural details, such as mitochondria cristae and microvesicles, could not be interpreted on TEM slides from FFPE tissues, as we noticed in our results. Here, we were only able to detect mitochondria presence, their number, localization, and partially their shape, but we could not see all the important details of their structure, as authors found in RCTs properly fixed for TEM analysis.[12]."

  1. "Please explain what scattered microvesicles in TEM mean. "

Response:

As we mentioned in the previous paragraph, macrovesicles were detected by TEM, and they were of similar consistency as mitochondria but smaller and scattered through the cytoplasm. The authors were not able to define their precise origin (references 12, 22). We noticed on our TEM images (despite poor resolution due to preparation from FFPE blocks) that some "microvesicles" could be detected since we compared the structure of mitochondria with the structure of smaller, paler vesicular structures of unknown origin in the same cell compartment.

  1. "In Discussion, it is described that all these results (IF and IHC) correlated with the TEM visualized mitochondria (Lines 299-300). In Figure 1, IHC and IF signals are weaker in pRCC than in chRCC and RO, but in Figure 2, it looks that density of mitochondria is higher in pRCC than in chRCC and RO. There is an inconsistency. "

Response:

Thank you very much for your concern. This inconsistency was apparent; however, the major reason was similar to the explanations addressed in your first and second comments. Indeed, Figure 1 showed low-grade pRCC (formerly pRCC, type 1) that seems to have slightly decreased mitochondria staining compared to high-grade pRCC (formerly pRCC, type 2), presented in Figure 2. For this reason, we prepared new Figures with both low- and high-grade pRCC. Thus, we hope that the present images are clear, as are our explanations in the Result and Discussion sections.

  1. "There are two kinds of cells, eosinophilic cells and pale cells in chRCC. I presume that authors analyzed eosinophilic cells. What are the results of mitochondria expression in pale cells?"

Response: Thank you very much for your suggestion. In the revised version of the manuscripts, new figures illustrate findings both in pale and eosionophilic cells in chRCC. These differences are also explained in the Results and Discussion.

  1. "In conclusion, it is described that IF using anti- mitochondria antibody can assist in fast and easy distinction of RO from chRCC and other eosinophilic variant of RCC with high accuracy. However, I can find little discussion that can result in a conclusion that IF is more useful than IHC. "

Response:

The last paragraphs of Discussion and Conclusion are adjusted according to your comments. We just wanted to underline that both IHC and IF could be useful as additional tools for precise differentiation of renal neoplasms. However, considering that IF is a very sensitive and specific type of immunostaining on fresh frozen sections, such as those used for ex tempore analysis, this IF staining could be beneficial compared to IHC. We sincerely hope that the current version of the Discussion and Conclusion is sufficiently improved to highlight the major study findings.

  1. In the Discussion, I recommend a table that specifies the differential characteristics of IHC and IF of these RCTs."

Response:

The table that specifies the differential characteristics of IHC and IF of these RCTs is included in the last paragraph of the Results, along with all the main findings in our study, illustrating the summary of data.

Round 2

Reviewer 2 Report

The authors have tried to defend the shortcomings in their work. I am not convinced with this revised work and clarifications provided. Unfortunately, the mitochondrial IF stainings are not looking organelle specific. These are ambiguous.  In this situation I had suggested for a second mitochondrial marker which again they have avoided. Lastly the TEM images are not suitable for these type of interpretation. 

Author Response

Reviewer 2:

The authors have tried to defend the shortcomings in their work. I am not convinced with this revised work and clarifications provided. Unfortunately, the mitochondrial IF stainings are not looking organelle specific. These are ambiguous.  In this situation I had suggested for a second mitochondrial marker which again they have avoided. Lastly the TEM images are not suitable for these type of interpretation. 

Dear reviewer,

Thank you very much for your kind comments and efforts to improve the quality of our manuscript. However, the main point and conclusion of our data consider the relevance of mitochondria staining and the usage of its different expression patterns as a fast tool for the differentiation of various renal cell tumours, especially those of benign biological behavior, from malignant tumours. In our work, TEM was only used to show that ultrastructural details, such as mitochondria, could be visible on a TEM FFPE sample, but without important details being destroyed due to improper fixation methods. Our TEM images are not suitable for detail interpretation, thus we already discussed it on Page 10 under the Discussion section: “It is also important to use an adequate fixation procedure for TEM, since all required fine ultrastructural details such as mitochondria cristae and microvesicles could not be interpreted on TEM slides form FFPE tissues, as we noticed in our results. Here, we were only able to detected mitochondria presence, their number, localization and partially their shape, but we could not see all mentioned important details of their structure, as authors found in RCTs properly fixed for TEM analysis [12]. Indeed, in order to use TEM for diagnostic purpose in uropathology, a piece of kidney tumor samples should be stored in adequate fixative.”

Moreover, there are several ways to investigate the specificity of antibody staining, especially when performed as IF. The first way is that during the evaluation of the IF slide performed with secondary goat anti-mouse IgG-Alexa 488 (green signal), there is a possibility to switch the filter from green to red or blue, and if the staining is specific, nothing will be visible on the red and blue filters, while non-specific staining will shine under all the filters used. This method was used in our study, and we did not detect nonspecific antibody binding. In addition, in this second revised version of the manuscript, we introduced the second method of antibody specificity control, considering IF negative controls, as described in Methods, Page 3: "Negative controls were performed in all experiments by omitting the first antibody." However, despite our efforts to capture images of IF negative controls, we failed to obtain any background staining, so the images were completely black, as shown below. In fact, we included IHC negative controls along with positive IHC and IF staining with the Prohibitin antibody. For the convenience of readers, we have included the images of the negative controls for all tumor types examined as an additional Figure 6. The same figure also includes the additional slides stained with Prohibitin monoclonal antibody (II -14-10) (MA5-12858) to confirm the conclusions. Prohibitin is an evolutionarily conserved protein located in the inner membrane of mitochondria. In contrast, initial staining with mitochondria-Ab2 (clone MTC02) recognizes non-glycosylated protein components of mitochondria. Thus, two different antibodies are specific for the detection of two different epitopes in mitochondria.

Negative IF controls for Mitochondria antibody

Considering that we kindly accepted all suggestions you gave us, and we introduce the second mitochondria antibody clone staining, we sincerely hope that our extensively revised manuscript with validation study, could be suitable for consideration for publication in Diagnostics Journal.

Reviewer 4 Report

As a rule, authors' point-to-point responses to the reviewer's comment is necessary for 2nd revision. 

Please upload your point-to-point responses along with the revised manuscript.

Author Response

Reviewer 4 (round 1):

1."In Table 1, distribution pattern of mitochondria staining is classified as coarse scanty, moderate scanty, or diffuse granular. Almost all ccRCC was coarse scanty, all papillary RCC (pRCC) was diffuse granular, all chRCC was moderate granular, and all RO was diffuse granular. Expression intensity is dichotomized in mild to moderate or strong. On the other hand, in the Discussion, it is also described that diffuse intense fine granular cytoplasmic staining in RO, intensive with coarse granularity in chRCC, and diffuse disperse granular in pRCC (Lines 287-288). As such, there are differences in description in between Table 1 and in Discussion, which are very confusing. I recommend description of IHC patterns united between them. n addition, description that diffuse uniform pattern throughout the whole cytoplasm (pRCC, low grade) and intense granular pattern apically accentuated (pRCC, high grade) (Lines 292-294) should also be revised."

Response:

In order to maintain the uniform terminology within the whole manuscript, we revised the text in the Results and Discussion (track changes) according to the nomenclature used in the Table 1 headings. With regard to your last comment,

Lines 292-294, we adjusted the description in the Results section:

"All pRCC samples expressed diffuse granular IHC staining patterns. Although the current 5th edition of the WHO classification of urinary and male genital tumours does not differ between pRCC as type 1 and type 2, we noticed slight differences in mitochondria antibody expression patterns depending on the morphology. Indeed, low-grade pRCC (defined as type 1 according to the previous edition of WHO classification) had a diffuse, uniform granular uniform pattern throughout the whole cytoplasm, while high-grade pRCC (defined as type 2 according to the previous edition of WHO classification) had a diffuse granular pattern with slight apical accentuation." In addition, the Discussion section was also modified. Thus, these sentences were deleted, but the changes were included and sufficiently explained in the Results.

  1. "Moderate granular pattern and diffuse granular pattern of mitochondria distribution cannot be well discriminated with Figure 1E, G, I. So, they should be replaced with images with higher magnification and higher resolution. In addition, diffuse uniform pattern throughout the whole cytoplasm (pRCC, low grade) and intense granular pattern apically accentuated (pRCC, high grade) should be presented with high resolution images."

Response:

Figure 1 is now modified with the higher magnification and higher resolution images. Moreover, the new Figure 3 contains two morphologically different types of pRCC. These two types were previously classified as pRCC type 1, and pRCC type 2 (WHO, 4th edition, 2016), reclassified as low and high grade pRCC in 5th edition of WHO classification (2022). Considering that pRCC type 2 morphologically contains more eosinophilic cells with abundant cytoplasm, contrary to pRCC type 1, the expression of anti-mitochondria antibody was slightly different, which we tried to explain with apically accentuated expression in high grade pRCC (pRCC, type 2).

3." In Table 1, a total number of cases of left and right sides is 94, fewer than 96."

Response:

In Table 1, the total number of RCTs divided by sides is indeed 94, because in two cases we did not receive clinical data. In the revised version of the manuscript, this information is included in the text "For two patients, there was no clinical data on tumour side; therefore, two out of 96 cases were excluded from statistical analysis".

4."In Table 1, please specify the nuclear grading system of ccRCC"

Response:

The nuclear grading system for ccRCC was used according to the WHO/ISUP scheme, and this is implemented in the Table 1 legend. Moreover, the following sentences are now implemented in Methodology Section 2.3: "All diagnosis and recommendations for nuclear grading were applied according to the new WHO classification, especially for pRCC, which are now classified as low and high grade. A nuclear grading system according to WHO/ISUP is also implemented."

  1. "In Figure 2, images of transmission electron microsope (TEM) should be explained in captions."

Response:

Under the Figures legend, we shortly described TEM finding, in investigated RCTs, while in the text of the Results, all details are sufficiently explained.

  1. "In Discussion, it is described that mitochondria in RO is larger than that of chRCC (Lines 257-258). However, as far as I see Fig. 2 L and O, it is not the case. "

Response:

In this part, we discussed literature data with further comparison to our findings without insights into the precise ultrastructural findings because our TEM results were based on FFPE tissue and, accordingly, were not properly fixed for precise TEM analysis. Therefore, we mentioned at the begging of this discussion part that these ultrastructural differences correspond to literature data and do not represent our findings. The following sentences are slightly adjusted according to your comments:"According to literature data, tumour cells of RO are filled with uniform and round mitochondria. They are closely packed in the cytoplasm, and their size is larger than those in chRCC. Beside mitochondria, scattered microvesicles are also present, and their number varies from cell to cell. They are either absent or very sparse, contrary to chRCC. In chRCC, the number and shape of mitochondria depend on the number of microvesicles. Their distribution can be diffuse or localized in the peripheral cytoplasm [12, 22]. However, TEM is not widely used for daily practice, considering the requirements of specially trained human resources (technicians and pathologists), along with the expensive equipment and high depreciation costs [23]. It is also important to use an adequate fixation procedure for TEM since all required fine ultrastructural details, such as mitochondria cristae and microvesicles, could not be interpreted on TEM slides from FFPE tissues, as we noticed in our results. Here, we were only able to detect mitochondria presence, their number, localization, and partially their shape, but we could not see all the important details of their structure, as authors found in RCTs properly fixed for TEM analysis.[12]."

  1. "Please explain what scattered microvesicles in TEM mean. "

Response:

As we mentioned in the previous paragraph, macrovesicles were detected by TEM, and they were of similar consistency as mitochondria but smaller and scattered through the cytoplasm. The authors were not able to define their precise origin (references 12, 22). We noticed on our TEM images (despite poor resolution due to preparation from FFPE blocks) that some "microvesicles" could be detected since we compared the structure of mitochondria with the structure of smaller, paler vesicular structures of unknown origin in the same cell compartment.

  1. "In Discussion, it is described that all these results (IF and IHC) correlated with the TEM visualized mitochondria (Lines 299-300). In Figure 1, IHC and IF signals are weaker in pRCC than in chRCC and RO, but in Figure 2, it looks that density of mitochondria is higher in pRCC than in chRCC and RO. There is an inconsistency. "

Response:

Thank you very much for your concern. This inconsistency was apparent; however, the major reason was similar to the explanations addressed in your first and second comments. Indeed, Figure 1 showed low-grade pRCC (formerly pRCC, type 1) that seems to have slightly decreased mitochondria staining compared to high-grade pRCC (formerly pRCC, type 2), presented in Figure 2. For this reason, we prepared new Figures with both low- and high-grade pRCC. Thus, we hope that the present images are clear, as are our explanations in the Result and Discussion sections.

  1. "There are two kinds of cells, eosinophilic cells and pale cells in chRCC. I presume that authors analyzed eosinophilic cells. What are the results of mitochondria expression in pale cells?"

Response: Thank you very much for your suggestion. In the revised version of the manuscripts, new figures illustrate findings both in pale and eosionophilic cells in chRCC. These differences are also explained in the Results and Discussion.

  1. "In conclusion, it is described that IF using anti- mitochondria antibody can assist in fast and easy distinction of RO from chRCC and other eosinophilic variant of RCC with high accuracy. However, I can find little discussion that can result in a conclusion that IF is more useful than IHC. "

Response:

The last paragraphs of Discussion and Conclusion are adjusted according to your comments. We just wanted to underline that both IHC and IF could be useful as additional tools for precise differentiation of renal neoplasms. However, considering that IF is a very sensitive and specific type of immunostaining on fresh frozen sections, such as those used for ex tempore analysis, this IF staining could be beneficial compared to IHC. We sincerely hope that the current version of the Discussion and Conclusion is sufficiently improved to highlight the major study findings.

  1. In the Discussion, I recommend a table that specifies the differential characteristics of IHC and IF of these RCTs."

Response:

The table that specifies the differential characteristics of IHC and IF of these RCTs is included in the last paragraph of the Results, along with all the main findings in our study, illustrating the summary of data.

Round 3

Reviewer 2 Report

Authors have clarified my concerns.

Author Response

Dear Reviewer, thank you very much for your suggestions during the submission of  our manuscript.

Best regards

Reviewer 4 Report

The authors have made satisfactory responses to my previous comments. After admitting that, I would like to add some comments to improve the manuscript.

1. (Lines 21-23 in Abstract) The main purpose of this study is to differentiate renal cell tumors (RCTs), especially between chromophobe renal cell carcinoma (chRCC) and renal oncocytoma (RO), by an ancillary immunohistochemical analysis of mitochodria expression. The abstract should focus on the association of mitochondria expression pattern with RCTs. Associations of genders, ages, tumor sides, tumor dimensions, and tumor stages with mitochodrial distribution pattern may be resulting from prevalence of specific RCT, and hence these information would be of low priority.

2. (Lines 21-22 in Abstract) “coarse granular staining” and “equally distributed”: Which pattern are these near, coarse scanty or moderate granular? As I pointed out previously, please unify the IHC pattern of mitochondria expression throughout the text.

3. (Line 50) cytokeratin (CK): Does this mean pan-cytokeratin? CK consists of many isoforms and please specify it.

4. (Line 258) chRCC showed coarse granular: Which pattern is this near, coarse scanty, moderate granular or diffuse granular? As I pointed out previously, please unify the IHC pattern of mitochondria expression throughout the text.

5. (Lines 364-365) In ccRCC, cytoplasmic staining was coarse scanty granular cytoplasmic in most cases.: Which pattern is this near, coarse scanty, moderate granular or diffuse granular? As I pointed out previously, please unify the IHC pattern of mitochondria expression throughout the text.

6. There are many grammatical errors.

(Line 26) presented the higher nuclear: presenting the higher nuclear.

(Line 47) clinical managenent tumor category: This phrase makes no sense.

(Line 75) renal cell tumors relay on: rely on.

(Line 130) The distribution pattern was defined as coarse, scanty: coarse scanty.

(Line 132) mild to moderate (1) and strong (3): Is it OK? Isn’t it “mild (1), moderate (2) and strong (3)”?

(Line 213 in Figure 1 caption) tubuli then distal: than.

(Line 219) randomly distributed Figure 2B,F: (Figures 2B,F)

(Line 220) Nevertheless, morphologically,: Morphologically,.

(Lines 234-236) Although the current ... type 1 and type 2,: This sentence makes no sense. Please rewrite so that we can understand.

(Line 322) there is no a single way: no single way.

(Line 344) we were only able to detected mitochondria: detect.

(Lines 365-366) However, we also detected the ccRCCs with higher nuclear grade and with eosinophilic cytoplasm with increased mitochondria expression: However, we also detected increased mitochondria expression in the ccRCCs with higher nuclear grade and with eosinophilic cytoplasm.

There are many grammatical errors and some sentences are difficult to understand. Although I pointed out some of them, English must be well edited.

Author Response

Response to Reviewer 4

The authors have made satisfactory responses to my previous comments. After admitting that, I would like to add some comments to improve the manuscript.

Comments to Abstract:

  1. (Lines 21-23 in Abstract) The main purpose of this study is to differentiate renal cell tumors (RCTs), especially between chromophobe renal cell carcinoma (chRCC) and renal oncocytoma (RO), by an ancillary immunohistochemical analysis of mitochodria expression. The abstract should focus on the association of mitochondria expression pattern with RCTs. Associations of genders, ages, tumor sides, tumor dimensions, and tumor stages with mitochodrial distribution pattern may be resulting from prevalence of specific RCT, and hence these information would be of low priority.
  2. (Lines 21-22 in Abstract) “coarse granular staining” and “equally distributed”: Which pattern are these near, coarse scanty or moderate granular? As I pointed out previously, please unify the IHC pattern of mitochondria expression throughout the text.

Response:

According to your kind suggestions we completely modified the Abstract, unifying the IHC pattern of mitochondria expression throughout the text, as well as highlighting the main purpose of this study which is differentiation renal cell tumors (RCTs), especially chromophobe renal cell carcinoma (chRCC) and renal oncocytoma (RO), by an ancillary immunohistochemical analysis of mitochondria expression:

" The precise differentiation of renal cell tumours (RCTs) is sometimes hard to achieve using standard imaging and histopathological methods, especially those with eosinophilic features. It has been suggested that the vast overabundance of mitochondria, is a well-known hallmark of eosinophilic cytoplasm, and could be a characteristic of distinct tumor types with opposing clinical outcomes. Thus, we intended to explore an association of mitochondrial distribution patterns in different RCTs, including 43 cell renal cell carcinomas (ccRCCs), 15 papillary renal cell carcinomas (pRCCs), 20 chromophobe renal cell carcinomas (chRCCs), 18 renal oncocytomas (ROs). Tumor samples were stained with two anti-mitochondria antibodies (Mitochondria antibody Ab-2, clone MTC02; Prohibitin, II-14-10, MA5-12858), applying immunohistochemistry and immunofluorescence, to define mitochondria distribution pattern (coarse scanty, moderate granular and diffuse granular). Our results revealed significantly different expression patterns among investigated RCTs (p<0.001). The majority of ccRCCs exhibited coarse scanty mitochondrial staining, while all chRCCs had moderate granular expression. Nevertheless, all ROs, all pRCCs, and two cases of ccRCCs presenting with higher nuclear grade and eosinophilic cytoplasm had diffuse granular mitochondria expression. Moreover, with increased distribution of mitochondria, the intensity of staining was higher (p<0.001). Here we present a strategy that utilizes fast and easy mitochondria detection to differentiate RO from chRCC, as well as other eosinophilic variants of RCC with high accuracy."

  1. (Line 50) cytokeratin (CK): Does this mean pan-cytokeratin? CK consists of many isoforms and please specify it.

Response:

Thank you very much for your suggestion. We made a correction, defining which CK isoform was mentioned in the text: ''Different immunohistochemical stains have been used to differentiate kidney tumours. It is known that both cytokeratin (CKAE1/AE3) and vimentin are positive in ccRCC and pRCC, while vimentin is negative in chRCC and RO."

  1. (Line 258) chRCC showed coarse granular: Which pattern is this near, coarse scanty, moderate granular or diffuse granular? As I pointed out previously, please unify the IHC pattern of mitochondria expression throughout the text.

Response:

Moderate granular cytoplasmic staining is the characteristic for chRCC, and we have tried to unify it throughout the whole manuscript. All changes are visible in the Word version of the revised manuscript with track changes.

  1. (Lines 364-365) In ccRCC, cytoplasmic staining was coarse scanty granular cytoplasmic in most cases.: Which pattern is this near, coarse scanty, moderate granular or diffuse granular? As I pointed out previously, please unify the IHC pattern of mitochondria expression throughout the text.

Response:

Coarse scanty cytoplasmic staining is the characteristic for ccRCC, and we have tried to unify it throughout the whole manuscript. All changes are visible in the Word version of the revised manuscript with track changes.

  1. There are many grammatical errors.

(Line 26) presented the higher nuclear: presenting the higher nuclear.

Response: According to your suggestions, corrections were made.

(Line 47) clinical managenent tumor category: This phrase makes no sense.

Response: According to your suggestions, we rewrote the sentence: '' These entities are not specific pathohistological diagnoses, and represent tumour category requiring a wide spectrum of additional research, influencing patients' treatment and outcome''

(Line 75) renal cell tumors relay on: rely on.

Response: According to your suggestions, corrections were made.

(Line 130) The distribution pattern was defined as coarse, scanty: coarse scanty.

Response: According to your suggestions, corrections were made.

(Line 132) mild to moderate (1) and strong (3): Is it OK? Isn’t it “mild (1), moderate (2) and strong (3)”?

Response: According to your suggestions, corrections were made. '' IHC stained slides were analyzed with regard to distribution patterns and intensity of mitochondria staining. The distribution patterns were defined as coarse scanty (score 1), moderate granular (score 2), and diffuse granular (score 3) cytoplasmic stainings, while intensity was mild (1), moderate (2), and strong (3). Finally, as a sum of these two scores, the IHC score was made.''

(Line 213 in Figure 1 caption) tubuli then distal: than.

Response: According to your suggestions, correction was made.

(Line 219) randomly distributed Figure 2B,F: (Figures 2B,F)

Response: According to your suggestions, corrections were made.

(Line 220) Nevertheless, morphologically,: Morphologically,.

Response: According to your suggestions, corrections were made.

(Lines 234-236) Although the current ... type 1 and type 2,: This sentence makes no sense. Please rewrite so that we can understand.

Response: According to your suggestions, corrections were made.

(Line 322) there is no a single way: no single way.

Response: According to your suggestions, corrections were made.

(Line 344) we were only able to detected mitochondria: detect.

Response: According to your suggestions, corrections were made.

(Lines 365-366) However, we also detected the ccRCCs with higher nuclear grade and with eosinophilic cytoplasm with increased mitochondria expression: However, we also detected increased mitochondria expression in the ccRCCs with higher nuclear grade and with eosinophilic cytoplasm.

Response: According to your kind suggestion, we implemented your sentence.